# A Biologically Inspired Visual Working Memory for Deep Networks

## Abstract

The ability to look multiple times through a series of pose-adjusted glimpses is fundamental to human vision. This critical faculty allows us to understand highly complex visual scenes. Short term memory plays an integral role in aggregating the information obtained from these glimpses and informing our interpretation of the scene. Computational models have attempted to address glimpsing and visual attention but have failed to incorporate the notion of memory. We introduce a novel, biologically inspired visual working memory architecture that we term the Hebb-Rosenblatt memory. We subsequently introduce a fully differentiable Short Term Attentive Working Memory model (STAWM) which uses transformational attention to learn a memory over each image it sees. The state of our Hebb-Rosenblatt memory is embedded in STAWM as the weights space of a layer. By projecting different queries through this layer we can obtain goal-oriented latent representations for tasks including classification and visual reconstruction. Our model obtains highly competitive classification performance on MNIST and CIFAR-10. As demonstrated through the CelebA dataset, to perform reconstruction the model learns to make a sequence of updates to a canvas which constitute a parts-based representation. Classification with the self supervised representation obtained from MNIST is shown to be in line with the state of the art models (none of which use a visual attention mechanism). Finally, we show that STAWM can be trained under the dual constraints of classification and reconstruction to provide an interpretable visual sketchpad which helps open the 'black-box' of deep learning.

## 1 Introduction

Much of the current effort and literature in deep learning focuses on performance from a statistical pattern recognition perspective. In contrast, we go back to a biological motivation and look to build a model that includes aspects of the human visual system. The eminent computational neuroscientist David Marr posited that vision is composed of stages which lead from a two dimensional input to a three dimensional contextual model with an established notion of object (Marr, 1982). This higher order model is built up in the visual working memory as a visual sketchpad which integrates notions of pattern and texture with a notion of pose (Baddeley and Hitch, 1974). Visual attention models often draw inspiration from some of these concepts and perform well at various tasks (Ablavatski et al., 2017; Ba et al., 2014; Gregor et al., 2015; Jaderberg et al., 2015; Mnih et al., 2014; Sønderby et al., 2015). Inspired by vision in nature, visual attention corresponds to adaptive filtering of the model input, typically, through the use of a glimpsing mechanism which allows the model to select a portion of the image to be processed at each step. Broadly speaking, visual attention models exist at the crux of two key challenges. The first is to separate notions of pose and object from visual features. The second is to effectively model long range dependencies over a sequence of observations.

Various models have been proposed and studied which hope to enable deep networks to construct a notion of pose. For example, transformational attention models learn an implicit representation of object pose by applying a series of transforms to an image (Jaderberg et al., 2015; Ablavatski et al., 2017). Other models such as Transformational Autoencoders and Capsule Networks harness an explicit understanding of positional relationships between objects (Hinton et al., 2011; Sabour et al., 2017). Short term memories have previously been studied as a way of improving the ability of Recurrent Neural Networks (RNNs) to learn long range dependencies. The ubiquitous Long Short-Term Memory (LSTM) network is perhaps the most commonly used example of such a model

(Hochreiter and Schmidhuber, 1997). More recently, the fast weights model, proposed by Ba et al. (2016) provides a way of imbuing recurrent networks with an ability to attend to the recent past.

From these approaches, it is evident that memory is a central requirement for any method which attempts to augment deep networks with the ability to attend to visual scenes. The core concept which underpins memory in neuroscience is synaptic plasticity, the notion that synaptic efficacy, the strength of a connection, changes as a result of experience (Purves et al., 1997). These changes occur at multiple time scales and, consequently, much of high level cognition can be explained in terms of the interplay between immediate, short and long term memories. An example of this can be found in vision, where each movement of our eyes requires an immediate contextual awareness and triggers a short term change. We then aggregate these changes to make meaningful observations over a long series of glimpses. Fast weights (Ba et al., 2016) draw inspiration from the Hebbian theory of learning (Hebb, 1949) which gives a framework for how this plasticity may occur. Furthermore, differentiable plasticity (Miconi et al., 2018) combines neural network weights with weights updated by a Hebbian rule to demonstrate that backpropagation can be used to learn a substrate over which the plastic network acts as a content-addressable memory.

In this paper, we propose augmenting transformational attention models with a visual working memory in order to move towards two key goals. Firstly, we wish to understand if visual attention and working memory provide more than just increased efficiency and enable functions that cannot otherwise be achieved. Secondly, we wish to understand and seek answers to some of the challenges faced when attempting to model such psychophysical concepts in deep networks. We demonstrate classification performance on MNIST (LeCun, 1998) and CIFAR-10 (Krizhevsky and Hinton, 2009) that is competitive with the state of the art and vastly superior to previous models of attention, demonstrating the value of a working memory. We then demonstrate that it is possible to learn this memory representation in an unsupervised manner by painting images, similar to the Deep Recurrent Attentive Writer (DRAW) network (Gregor et al., 2015). Using this representation, we demonstrate competitive classification performance on MNIST with self supervised features. Furthermore, we demonstrate that the model can learn a disentangled space over the images in CelebA (Liu et al., 2015), shedding light on some of the higher order functions that are enabled by visual attention. Finally, we show that the model can perform multiple tasks in parallel and how a visual sketchpad can be used to produce interpretable classifiers.

## 2 Related Work and the Psychology of Visual Memory

In this section we discuss related work on visual attention and relevant concepts from psychology and neuroscience which have motivated our approach. We will use the terms motivation and inspiration interchangeably throughout the paper to capture the notion that our decisions have been influenced by atypical factors. This is necessary as there are several facets to our approach which may seem nonsensical outside of a biological context.

Attention in deep models can be broadly split into two types: hard attention and soft attention. In hard attention a non-differentiable step is performed to extract the desired pixels from the image to be used in later operations. Conversely, in soft attention, differentiable interpolation is used. The training mechanism differs between the two approaches. Early hard attention models such as the Recurrent Attention Model (RAM) and the Deep Recurrent Attention Model (DRAM) used the REINFORCE algorithm to learn an attention policy over non-differentiable glimpses (Mnih et al., 2014; Ba et al., 2014). More recent architectures such as Spatial Transformer Networks (STNs) (Jaderberg et al., 2015), Recurrent STNs (Sønderby et al., 2015) and the Enriched DRAM (EDRAM) (Ablavatski et al., 2017) use soft attention and are trained end to end with backpropagation. The DRAM model and its derivatives use a two layer LSTM to learn the attention policy. The first layer is intended to aggregate information from the glimpses and the second layer to observe this information in the context of the whole image and decide where to look next. The attention models described thus far predominantly focus on single and multi-object classification on the MNIST and Street View House Numbers (Goodfellow et al., 2013) datasets respectively. Conversely, the DRAW network from Gregor et al. (2015) is a fully differentiable spatial attention model that can make a sequence of updates to a canvas in order to draw the input image. Here, the canvas acts as a type of working memory which is constrained to also be the output of the network.

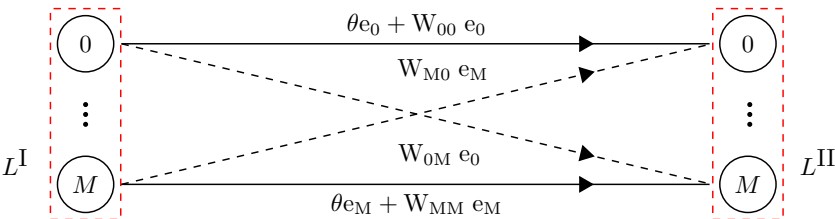

Figure 1: The two layer Hebb-Rosenblatt memory architecture. This is a novel, differentiable module inspired by Rosenblatts perceptron that can be used to 'learn' a Hebbian style memory representation over a sequence and can subsequently be projected through to obtain a latent space.

A common theme to these models is the explicit separation of pose and feature information. The Two Streams Hypothesis, suggests that in human vision there exists a 'what' pathway and a 'where' pathway (Goodale and Milner, 1992). The 'what' pathway is concerned with recognition and classification tasks and the 'where' pathway is concerned with spatial attention and awareness. In the DRAM model, a multiplicative interaction between the pose and feature information, first proposed in Larochelle and Hinton (2010), is used to emulate this. The capacity for humans to understand pose has been studied extensively. Specifically, there is the notion that we are endowed with two methods of spatial understanding. The first is the ability to infer pose from visual cues and the second is the knowledge we have of our own pose (Gibson, 1950).

Memory is generally split into three temporal categories, immediate memory, short term (working) memory and long term memory (Purves et al., 1997). Weights learned through error correction procedures in a neural network are analogous to a long term memory in that they change gradually over the course of the models existence. We can go further to suggest that the activation captured in the hidden state of a recurrent network corresponds to an immediate memory. There is, however, a missing component in current deep architectures, the working memory, which we study here. We will later draw inspiration from the Baddeley model of working memory (Baddeley and Hitch, 1974) which includes a visual sketchpad that allows individuals to momentarily create and revisit a mental image that is pertinent to the task at hand. This sketchpad is the integration of 'what' and 'where' information that is inferred from context and/or egomotion.

## 3 An Associative Visual Working Memory

Here we will outline our approach to augmenting models of visual attention with a working memory in order to seek a deeper understanding of the value of visual attention. In Section 3.1 we will detail the 'plastic' Hebb-Rosenblatt network and update policy that comprise our visual memory. Then in Section 3.2 we will describe the STAWM model of visual attention which is used to build up the memory representation for each image. The key advantage of this approach that will form the basis of our experimentation lies in the ability to project multiple query vectors through the memory and obtain different, goal-oriented, latent representations for one set of weights. In Section 3.3 we will discuss the different model 'heads' which use these spaces to perform different tasks.

### 3.1 Hebb-Rosenblatt Redux

We will now draw inspiration from various models of plasticity to derive a differentiable learning rule that can be used to iteratively update the weights of a memory space, during the forward pass of a deep network. Early neural network models used learning rules derived from the Hebbian notion of synaptic plasticity (Hebb, 1949; Block, 1962; Block et al., 1962; Rosenblatt, 1962). The phrase 'neurons that fire together wire together' captures this well. If two neurons activate for the same input, we increase the synaptic weight between them; otherwise we decrease. In a spiking neural network this activation is binary, *on* or *off*. However, we can obtain a continuous activation by integrating over the spike train in a given window $\Delta t$. Although the information in biological networks is typically seen as being encoded by the timing of activations and not their number, this spike train integral can approximate it to a reasonable degree (Dayan and Abbott, 2001). Using this, we can create

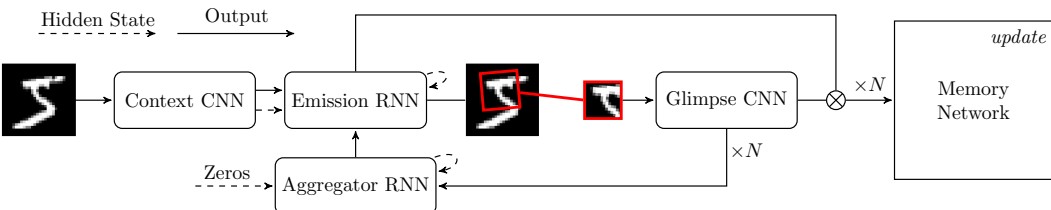

Figure 2: The attentional working memory model which produces an *update* to the memory. See Section 3.2 for a description of the structure and function of each module.

a novel short-term memory unit that can be integrated with other networks that are trained with backpropagation.

Consider the two layer network shown in Figure 1 with an input layer $L^{\mathrm{I}}$ and output layer $L^{\mathrm{II}}$, each of size $M$ with weights $\mathbf{W} \in \mathbb{R}^{M \times M}$. For a signal vector $\mathbf{e} \in \mathbb{R}^M$, propagating through $L^{\mathrm{I}}$, $L^{\mathrm{II}}$ will activate according to some activation function $\phi$ of the projection $\mathbf{We}$. The increase in the synaptic weight will be proportional to the product of the activations in the input and output neurons. We can model this with the outer product (denoted $\otimes$) to obtain the increment expression

$$\mathbf{W}(t+1) = \mathbf{W}(t) + \eta[\mathbf{e} \otimes \phi(\mathbf{W}(t)\mathbf{e})] - \delta[\mathbf{W}(t)] \ , \tag{1}$$

where at each step, we reduce the weights matrix by some decay rate $\delta$ and apply the increment with some learning rate $\eta$. This rule allows for the memory to make associative observations over the sequence of states such that salient information will become increasingly prevalent in the latent space. However, if we initialize the weights to zero, the increment term will always be zero and nothing can be learned. We could perhaps consider a different initialisation such as a Gaussian but this would be analogous to implanting a false memory, which may be undesirable. Instead, a solution is offered in the early multi layer perceptron models of Rosenblatt (1962). Here, neurons in the first layer transmit a fixed amount, $\theta$, of their input to a second layer counterpart as well as the weighted signal, as seen in Figure 1. Combining both terms we obtain the final, biologically inspired learning rule that we will herein refer to as the Hebb-Rosenblatt rule,

$$\mathbf{W}(t+1) = \mathbf{W}(t) + \eta[\mathbf{e} \otimes \phi(\mathbf{W}(t)\mathbf{e} + \theta\mathbf{e})] - \delta[\mathbf{W}(t)] \ . \tag{2}$$

Note that if we remove the projection term ($\mathbf{We}$) and set $\theta = 0$ and $\phi(x) = x$ we obtain the term used in the fast weights model of Ba et al. (2016) which also shares some similarities with the rules considered in Miconi et al. (2018). Analytically, this learning rule can be seen to develop high values for features which are consistently active and low values elsewhere. In this way the memory can make observations about the associations between salient features from a sequence of images.

## 3.2 The Short Term Attentive Working Memory Model (STAWM)

Here we describe in detail the STAWM model shown in Figure 2. This is an attention model that allows for a sequence of sub-images to be extracted from the input so that we can iteratively learn a memory representation from a single image. STAWM is based on the Deep Recurrent Attention Model (DRAM) and uses components from Spatial Transformer Networks (STNs) and the Enriched DRAM (EDRAM) (Ba et al., 2014; Jaderberg et al., 2015; Ablavatski et al., 2017). The design is intended to be comparable with previous models of visual attention whilst preserving proximity to the discussed psychological concepts, allowing us to achieve the goals set out in the introduction. At the core of the model, a two layer RNN defines an attention policy over the input image. As with EDRAM, each glimpse is parameterised by an affine matrix, $\mathbf{A} \in \mathbb{R}^{3 \times 2}$, which is sampled from the output of the RNN. At each step, $\mathbf{A}$ is used to construct a flow field that is interpolated over the image to obtain a fixed size glimpse in a process denoted as the glimpse transform, $t_{\mathbf{A}} : \mathbb{R}^{H_i \times W_i} \rightarrow \mathbb{R}^{H_g \times W_g}$, where $H_g \times W_g$ and $H_i \times W_i$ are the sizes of the glimpse and image respectively. Typically the glimpse is a square of size $S_g$ such that $H_g = W_g = S_g$. Features obtained from the glimpse are then combined with the location features and used to update the memory with Equation 2.

**Context and Glimpse CNNs:** The context and glimpse CNNs are used to obtain features from the image and the glimpses respectively. The context CNN is given the full input image and expected to

establish the contextual information required when making decisions about where to glimpse. The precise CNN architecture depends on the dataset used and can be found in Appendix D. We avoid pooling as we wish for the final feature representation to preserve as much spatial information as possible. Output from the context CNN is used as the initial input and initial hidden state for the emission RNN. From each glimpse we extract features with the glimpse CNN which typically has a similar structure to the context CNN.

**Aggregator and Emission RNNs:** The aggregator and emission RNNs, shown in Figure 2, formulate the glimpse policy over the input image. The aggregator RNN is intended to collect information from the series of glimpses in its hidden state which is initialised to zero. The emission RNN takes this aggregate of knowledge about the image and an initial hidden state from the context network to inform subsequent glimpses. By initialising the hidden states in this way, we expect the model to learn an attention policy which is motivated by the difference between what has been seen so far and the total available information. We use LSTM units for both networks because of their stable learning dynamics (Hochreiter and Schmidhuber, 1997), both with the same hidden size. As these networks have the same size they can be conveniently implemented as a two layer LSTM. We use two fully connected layers to transpose the output down to the six dimensions of $\mathbf{A}$ for each glimpse. The last of these layers has the weights initialised to zero and the biases initialised to the affine identity matrix as with STNs.

**Memory Network:** The memory network takes the output from a multiplicative 'what, where' pathway and passes it through a square Hebb-Rosenblatt memory with weights $\mathbf{W} \in \mathbb{R}^{M \times M}$, where $M$ is the memory size. The 'what' and 'where' pathways are fully connected layers which project the glimpse features ('what') and the RNN output ('where') to the memory size. We then take the elementwise product of these two features to obtain the input to the memory, $\mathbf{e} \in \mathbb{R}^M$. We can think of the memory network as being in one of three states at any point in time, these are: *update*, *intermediate* and *terminal*. The *update* state of the memory is a dynamic state where any signal which propagates through it will trigger an update to the weights using the rule in Equation 2. In STAWM, this update will happen $N$ times per image, once for each glimpse. Each of the three learning rates ($\delta$, $\eta$ and $\theta$) are hyper-parameters of the model. These can be made learnable to allow for the model to trade-off between information recall, associative observation and the individual glimpse. However, the stability of the memory model is closely bound to the choices of learning rate and so we derive necessary initial conditions in Appendix A.

For the *intermediate* and *terminal* states, no update is made to the Hebb-Rosenblatt memory for signals that are projected through. In the *intermediate* state we observe the memory at some point during the course of the attention policy. Conversely, in the *terminal* state we observe the fixed, final value for the memory after all $N$ glimpses have been made. We can use the *intermediate* or *terminal* states to observe the latent space of our model conditioned on some query vector. That is, at different points during the glimpse sequence, different query vectors can be projected through the memory to obtain different latent representations. For a self-supervised setting we can fix the weights of STAWM so that the memory inputs cannot be changed by the optimizer.

### 3.3 USING THE MEMORY

We now have an architecture that can be used to build up or 'learn' a Hebb-Rosenblatt memory over an image. Note that when projecting through the memory, we do not include $\theta\mathbf{e}$ in the output. This is so that it is not possible for the memory to simply learn the identity function. We use the output from the context CNN as a base that is projected into different queries for the different aims of the network. We do this using linear layers, the biases of which can learn a static representation which is then modulated by the weighted image context. We detach the gradient of the image context so that the context network is not polluted by divergent updates. In this section we will characterise the two network 'heads' which make use of latent spaces derived from the memory to perform the tasks of classification and drawing. We will also go on to discuss ways of constraining the glimpse sub-spaces in a variational setting.

**Classification:** For the task of classification, the query vector is projected through the memory in exactly the same fashion as a linear layer to derive a latent vector. This latent representation is then passed through a single classification layer which projects from the memory space down to the

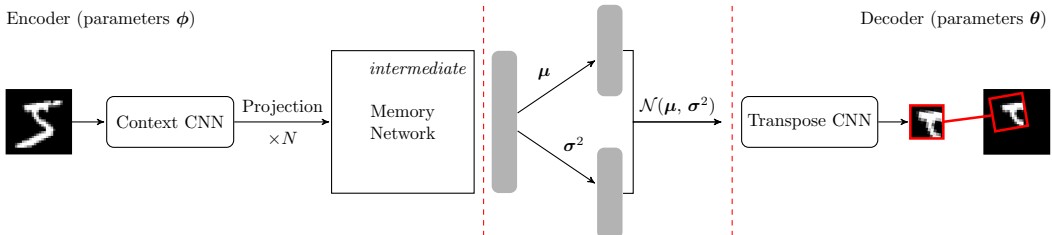

Figure 3: The classification model which uses the *terminal* state of the memory.

Figure 4: The drawing model which uses *intermediate* states of the memory.

number of classes as shown in Figure 3. A softmax is applied to the network output and the entire model (including the attenion mechanism) is trained by backpropagation to minimise the categorical cross entropy between the network predictions and the ground truth targets.

**Learning to Draw:** To construct a visual sketchpad we will use a novel approach to perform the same task as the DRAW network with the auxiliary model in Figure 4. Our approach differs from DRAW in that we have a memory that is independent from the canvas and can represent more than just the contents of the reconstruction. However, it will be seen that an important consequence of this is that it is not clear how to use STAWM as a generative model in the event that some of the skecthes are co-dependent. The drawing model uses each intermediate state of the memory to query a latent space and compute an update, $\mathbf{U} \in \mathbb{R}^{H_g \times W_g}$, to the canvas, $\mathbf{C} \in \mathbb{R}^{H_i \times W_i}$, that is made after each glimpse. Computing the update or sketch is straightforward, we simply use a transpose convolutional network ('Transpose CNN') with the same structure as the glimpse CNN in reverse. However, as the features were observed under the glimpse transform, $t_{\mathbf{A}}$, we allow the emission network to further output the parameters, $\mathbf{A}^{-1} \in \mathbb{R}^{3 \times 2}$, of an inverse transform, $t_{\mathbf{A}^{-1}} : \mathbb{R}^{H_g \times W_g} \to \mathbb{R}^{H_i \times W_i}$, at the same time. The sketch will be warped according to $t_{\mathbf{A}^{-1}}$ before it is added to the canvas. To add the sketch to the canvas there are a few options. We will consider two possibilities here: the addition method and the Bernoulli method.

The addition method is to simply add each update to the canvas matrix and finally apply a sigmoid after the glimpse sequence has completed to obtain pixel values. This gives an expression for the final canvas

$$\mathbf{C}_N = \mathrm{S}\left(\mathbf{C}_0 + \sum_{n=0}^{N} t_{\mathbf{A}^{-1}}(\mathbf{U}_m)\right) \quad, \tag{3}$$

where $\mathbf{C}_0 \in \{-6\}^{H_i \times W_i}$, S is the sigmoid function and $N$ is the total number of glimpses. We set $\mathbf{C}_0$ to $-6$ so that when no additions are made the canvas is black and not grey. The virtue of this method is its simplicity, however, for complex reconstructions, overlapping sketches will be required to counteract each other such that they may not be viewable independently in an interprettable manner.

An alternative approach, the Bernoulli method, could help to prevent these issues. Ideally, we would like the additions to the canvas to be as close as possible to painting in real life. In such a case, each brush stroke replaces what previously existed underneath it, which is dramatically different to the effect of the addition method. In order to achieve the desired replacement effect we can allow the model to mask out areas of the canvas before the addition is made. We therefore add an extra channel to the output of the transpose CNN, the alpha channel. This mask, $\mathbf{P} \in \mathbb{R}^{H_g \times W_g}$, is warped by $t_{\mathbf{A}^{-1}}$ as with the rest of the sketch. A sigmoid is applied to the output from the transpose CNN so that the mask contains values close to one where replacement should occur and close to zero elsewhere. Ideally, the mask values would be precisely zero or one. To achieve this, we could take

$\mathbf{P}$ as the probabilities of a Bernoulli distribution and then draw $\mathbf{B} \sim Bern(t_{\mathbf{A}^{-1}}(\mathrm{S}(\mathbf{P})))$. However, the Bernoulli distribution cannot be sampled in a differentiable manner. We will therefore use an approximation, the Gumbel-Softmax (Jang et al., 2016) or Concrete (Maddison et al., 2016) distribution, which can be differentiably sampled using the reparameterization trick. The Concrete distribution is modulated with the temperature parameter, $\tau$, such that $\lim_{\tau \to 0} Concrete(p, \tau) = Bern(p)$. We can then construct the canvas iteratively with the expression

$$\mathbf{C}_N = \mathbf{C}_{N-1} \odot (1 - \mathbf{B}) + t_{\mathbf{A}^{-1}}(\mathrm{S}(\mathbf{U}_N)) \odot \mathbf{B} \ , \ \mathbf{B} \sim Concrete(t_{\mathbf{A}^{-1}}(\mathrm{S}(\mathbf{P})), \tau) \ , \quad (4)$$

where $\mathbf{C}_0 \in \{0\}^{H_i \times W_i}$ and $\odot$ is the elementwise multiplication operator. Note that this is a simplified *over* operator from alpha compositing where we assume that objects already drawn have an alpha value of one (Porter and Duff, 1984).

**Constraining Glimpse Sub-spaces:** A common approach in reconstructive models is to model the latent space as a distribution whose shape over a mini-batch is constrained using a Kullback-Liebler divergence against a (typically Gaussian) prior distribution. We employ this variational approach as shown in Figure 4 where, as with Variational Auto-Encoders (VAEs), the latent space is modelled as the variance ($\boldsymbol{\sigma}^2$) and mean ($\boldsymbol{\mu}$) of a multivariate Gaussian with $K$ components and diagonal covariance (Kingma and Welling, 2013). For input $\mathbf{x} \in \mathbb{R}^{H_i \times W_i}$ and some sample $\mathbf{z} \in \mathbb{R}^K$ from the latent space, the $\beta$-VAE (Higgins et al., 2016) uses the objective

$$\mathcal{L}(\theta, \phi; \mathbf{x}, \mathbf{z}, \beta) = \mathbb{E}_{q_\phi(\mathbf{z} \mid \mathbf{x})} \left[ \log p_\theta(\mathbf{x} \mid \mathbf{z}) \right] - \beta D_{\mathrm{KL}} \left( q_\phi(\mathbf{z} \mid \mathbf{x}) \| p(\mathbf{z}) \right) \ . \quad (5)$$

For our model, we do not have a single latent space but a sequence of glimpse sub-spaces $G = (\mathbf{g}_n)_{n=0}^N$, $\mathbf{g}_n \in \mathbb{R}^K$. For the addition method, we can simply concatenate the sub-spaces to obtain $\mathbf{z} \in \mathbb{R}^{N \times K}$ and use the divergence term above. In the Bernoulli method, however, we can derive a more appropriate objective by acknowledging that outputs from the decoder are conditioned on the joint distribution of the glimpse sub-spaces. In this case, assuming that elements of $G$ are conditionally independent, following from the derivation given in Appendix B, we have

$$\mathcal{L}(\theta, \phi; \mathbf{x}, G, \beta) = \mathbb{E}_{q_\phi(G \mid \mathbf{x})} \left[ \log p_\theta(\mathbf{x} \mid G) \right] - \beta \sum_{n=0}^N D_{\mathrm{KL}} \left( q_\phi(\mathbf{g}_n \mid \mathbf{x}) \| p(\mathbf{g}_n) \right) \ . \quad (6)$$

## 4 Experiments

In this section we discuss the results that have been obtained using the STAWM model. The training scheme and specific architecture details differ for each setting, for full details see Appendix D. For the memory we use a ReLU6 activation, $y = \min(\max(x, 0), 6)$ (Krizhevsky and Hinton, 2010) on both the input and output layers. Following the analysis in Appendix A, we initialise the memory learning rates $\delta$, $\eta$ and $\theta$ to 0.2, 0.4 and 0.5 respectively. For some of the experiments these are learnable and are updated by the optimiser during training. Our code is implemented in PyTorch (Paszke et al., 2017) with torchbearer (Harris et al., 2018) and can be found at `https://github.com/iclr2019-anon/STAWM`. Examples in all figures have not been cherry picked.

### 4.1 Classification

We first perform classification experiments on handwritten digits from the MNIST dataset (LeCun, 1998) using the model in Figure 3. We perform some experiments with $S_g = 8$ in order to be comparable to previous results in the literature. We also perform experiments with $S_g = 28$ to see whether attention can be used to learn a positional manifold over an image by subjecting it to different transforms. The MNIST results are reported in Table 1 and show that STAWM obtains superior classification performance on MNIST compared to the RAM model. It can also be seen that the over-complete strategy obtains performance that is competitive with the state of the art of around $0.25\%$ for a single model (Sabour et al., 2017), with the best STAWM model from the 5 trials obtaining a test error of $0.31\%$. This suggests an alternative view of visual attention as enabling the model to learn a more powerful representation of the image. We experimented with classification on CIFAR-10 (Krizhevsky and Hinton, 2009) but found that the choice of glimpse CNN was the dominating factor in performance. For example, using MobileNetV2 (Sandler et al., 2018) as the glimpse CNN we obtained a single run accuracy of $93.05\%$.

Table 1: Classification performance of our model on the MNIST and CIFAR-10 datasets. Mean and standard deviation reported from 5 trials.

**(a) MNIST Supervised**

| Model | Error |
|---|---|
| RAM, $S_g = 8$, $N = 5$ | $1.55\%$ |
| RAM, $S_g = 8$, $N = 6$ | $1.29\%$ |
| RAM, $S_g = 8$, $N = 7$ | $1.47\%$ |
| STAWM, $S_g = 8$, $N = 8$ | $0.41\%_{\pm 0.03}$ |
| STAWM, $S_g = 28$, $N = 10$ | $0.35\%_{\pm 0.02}$ |

**(b) MNIST Self-supervised**

| Model | Error |
|---|---|
| DBM, Dropout (Srivastava et al., 2014) | $0.79\%$ |
| Adversarial (Goodfellow et al., 2014) | $0.78\%$ |
| Virtual Adversarial (Miyato et al., 2015) | $0.64\%$ |
| Ladder (Rasmus et al., 2015) | $0.57\%$ |
| STAWM, $S_g = 6$, $N = 12$ | $0.77\%$ |

**(c) CIFAR-10 Self-supervised**

| Model | Error |
|---|---|
| Baseline $\beta$-VAE | $63.44\%_{\pm 0.31}$ |
| STAWM, $S_g = 16$, $N = 8$ | $55.40\%_{\pm 0.63}$ |

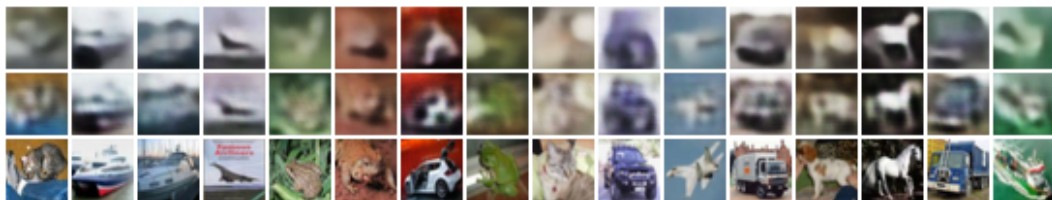

Figure 5: CIFAR-10 reconstructions for the baseline $\beta$-VAE and the drawing model with $N = 8$, $S_g = 16$. Top: baseline results. Middle: drawn results. Bottom: target images. Best viewed in colour.

## 4.2 DRAWING - ADDITION

Following experimentation with MNIST we observed that there are three valid ways to draw using the addition method. First, the model could simply compress the image into a square equal to the glimpse size and decompress later. Second, the model could learn to trace lines such that all of the notion of object is contained in the pose information instead of the features. Finally, the model could learn a pose invariant, parts based representation as we originally intended. The most significant control of this behaviour is the glimpse size. At $S_g = 8$ or above, enough information can be stored to make a simple compression the most successful option. Conversely, at $S_g = 4$ or below, the model is forced to simply draw lines. In light of these observations, we use $N = 12$ with $S_g = 6$ to obtain an appropriate balance. Sample update sequences for these models are shown in Figure 8 in Appendix C. For $S_g = 6$ it can be seen that the model has learned a repeatable parts-based representation which it uses to draw images. In this way the model has learned an implicit notion of class.

We also experimented with painting images in CIFAR-10. To establish a baseline we also show reconstructions from a reimplementation of $\beta$-VAE (Higgins et al., 2016). To be as fair as possible, our baseline uses the same CNN architecture and latent space size as STAWM. This is still only an indicative comparison as the two models operate in fundamentally different ways. Autoencoding CIFAR-10 is a much harder task due to the large diversity in the training set for relatively few images. However, our model significantly outperforms the baseline with a terminal mean squared error of $0.0083_{\pm 0.0006}$ vs $0.0113_{\pm 0.0001}$ for the VAE. On inspection of the glimpse sequence given in Appendix C we can see that although STAWM has not learned the kind of parts-based representation we had hoped for, it has learned to scan the image vertically and produce a series of slices. We again experimented with different glimpse sizes and found that we were unable to induce the desired behaviour. The reason for this seems clear; any 'edges' in the output where one sketch overlaps another would have values that are scaled away from the target, resulting in a constant pressure for the sketch space to expand. This is not the case in MNIST, where overlapping values will only saturate the sigmoid.

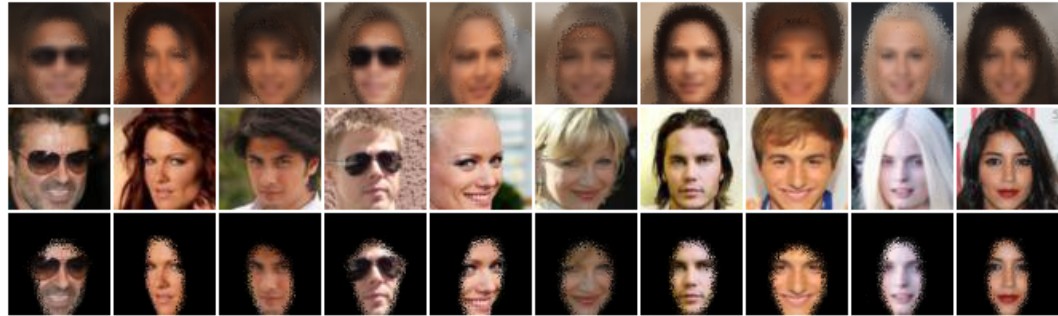

Figure 6: Output from the drawing model for CelebA with $N = 8$, $S_g = 32$. Top: the drawn results. Middle: the target images. Bottom: the learned mask from the last glimpse, pointwise multiplied with the target image. Best viewed in colour.

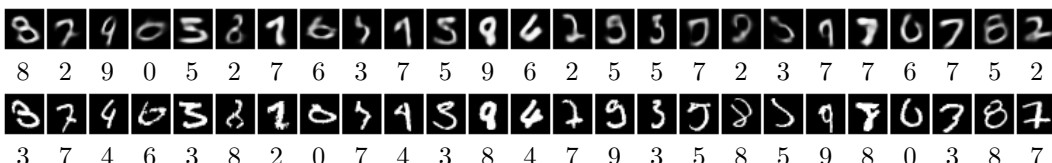

8 2 9 0 5 2 7 6 3 7 5 9 6 2 5 5 7 2 3 7 7 6 7 5 2

3 7 4 6 3 8 2 0 7 4 3 8 4 7 9 3 5 8 5 9 8 0 3 8 7

Figure 7: Top: sketchpad results and associated predictions for a sample of misclassifications. Bottom: associated input images and target classes.

**Self-Supervised Classification:** We can fix the weights of the STAWM model learned from the drawing setting and add the classification head to learn to classify using the self-supervised features. In this case, the only learnable parameters are the weights of the two linear layers and the learning rates for the memory construction. This allows the model to place greater emphasis on previous glimpses to aid classification. As can be seen from Table 1, we obtain performance on the self-supervised MNIST features that is competitive with the state of the art models, none of which use an attention mechanism. The self-supervised performance on CIFAR-10 is less competitive with the state of the art, but does show a clear improvement over the results obtained using the baseline VAE.

### 4.3 DRAWING - BERNOULLI

As discussed previously, the canvas for the drawing model can be constructed using the Bernoulli method given in Section 3.3. In this setting we trained on the CelebA dataset (Liu et al., 2015) which contains a large number of pictures of celebrity faces and add the KL divergence for the joint distribution of the glimpse spaces with a Gaussian prior given in Equation 25 in Appendix B. An advantage of the Bernoulli method is that we sample an explicit mask for the regions drawn at each step. If interesting regions are learned, we can use this as an unsupervised segmentation mask. Figure 6 shows the result of the drawing model on CelebA along with the target image and the learned mask from the final glimpse elementwise multiplied with the ground truth. Here, STAWM has learned to separate the salient face region from the background in an unsupervised setting. Analytically, the KL term we have derived will ask for the subsequent glimpse spaces to transition smoothly as the image is observed. Coupled with the fact that the memory can only build up a representation over the sequence, it follows that the model will learn to sketch increasingly complex areas with each new glimpse. This is precisely the case, as seen from the sketch sequence, Figure 10 in Appendix C.

### 4.4 VISUAL SKETCHPAD

One of the interesting properties of the STAWM model is the ability to project different things through the memory to obtain different views on the latent space. We can therefore use both the drawing network and the classification network in tandem by simply summing the losses. We scale each loss so that one does not take precedence over the other. We also must be careful to avoid overfitting so that the drawing is a good reflection of the classification space. For this experiment, with $S_g = 8$, the

terminal classification error for the model is $1.0\%$. We show the terminal state of the canvas for a sample of misclassifications in Figure 7. Here, the drawing gives an interesting reflection of what the model 'sees' and in many cases the drawn result looks closer to the predicted class than to the target. For example, in the rightmost image, the model has drawn and predicted a '2' despite attending to a '7'. This advantage of memories constructed as the weights of a layer, coupled with their ability to perform different tasks well with shared information, is evident and a clear point for further research.

## 5    CONCLUSIONS AND FUTURE WORK

In this paper we have described a novel, biologically motivated short term attentive working memory model (STAWM) which demonstrates impressive results on a series of tasks and makes a strong case for further study of short term memories in deep networks. As well as demonstrating competitive classification results on MNIST and CIFAR-10, we have shown that the core model can be used for image reconstruction and for disentangling foreground from background in an unsupervised setting with CelebA. Finally, we have given a concrete example of how a model augmented with a visual sketchpad can 'describe what it sees' in a way that is naturally interpretable for humans. It is easy to see how similar systems could be used in future technologies to help open the 'black-box' and understand why a decision was made, through the eyes of the model that made it. Furthermore, we have explored the notion that building up a memory representation over an attention policy, coupled with a smooth changing latent space can result in a movement from simple to complex regions of a scene. This perhaps gives us some insight into how humans learn to attend to their environment when endowed with a highly capable visual memory. Future work will look to see if variants of this model can be used to good effect on higher resolution images. Experimentation and analysis should also be done to further understand the dynamics of the Hebb-Rosenblatt memory and the representation it learns. We further intend to investigate if the memory model can be used for other applications such as fusion of features from multi-modal inputs. Finally, we will look further into the relationship between visual memories and saliency.

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

## A   STABILISING THE MEMORY

The working memory model described in the paper exhibits a potential issue with stability. It is possible for the gradient to explode, causing damaging updates which halt learning and from which the model cannot recover. In this section we will briefly demonstrate some properties of the learning rule in Equation 2 and derive some conditions under which the dynamics are stable. We will broadly follow the method of Block et al. (1962) with minor alterations for our approach. We use the terms stimuli and response to represent input to and output from a neuron respectively. We will consider a sequence of $H_g \times W_g$ glimpse stimuli, $G = (g_n)_{n=0}^N, \; g_n \in \mathbb{R}^{H_g \times W_g}$, presented to $L^{\mathrm{I}}$ when attending to a single image.

First, let $e_\mu^{(i)} = \phi^{\mathrm{I}}(x^{(i)})$ represent the response of some neuron $\mu$ in $L^{\mathrm{I}}$ as a nonlinear function $\phi^{\mathrm{I}}$ of its input $x^{(i)}$ for some stimulus $i$. Next, we define $m_{ij}^{\mathrm{I}} = \sum_\mu e_\mu^{(i)} e_\mu^{(j)}$ as the sum of the product of first layer responses to both stimuli $i$ and $j$. Here we introduce the first constraint: $m_{ij}^{\mathrm{I}}$ does not change over time. This is true if the weights of the feature transform do not change when attending to a single image and if the feature $x^{(i)}$ does not depend on the output of a recurrent network. Now, we define $\beta_\nu^{(i)}$ as the identity input to some $\nu \in L^{\mathrm{II}}$. We then define $\gamma_\nu^{(i)}$, the projection of $\mathbf{e}^{(i)}$ through the weights matrix $\mathbf{W} \in \mathbb{R}^{M \times M}$. The total input to $\nu$ if stimulus $i$ is presented to $L^{\mathrm{I}}$ at time $t$ is the sum of these two terms given in Equation 9.

$$\beta_\nu^{(i)} = \theta e_\nu^{(i)} \tag{7}$$

$$\gamma_\nu^{(i)}(t) = \sum_\mu \mathrm{W}_{\mu\nu}(t) \mathrm{e}_\mu^{(i)} \tag{8}$$

$$a_\nu^{(i)}(t) = \beta_\nu^{(i)} + \gamma_\nu^{(i)}(t) \tag{9}$$

Suppose that a stimulus $i$ is presented at time $t_0$ for some period $\Delta t$. The subsequent change in the weight, $w_{\mu\nu}$, of some connection is defined in Equation 10, where $\phi^{\mathrm{II}}$ is a nonlinear activation function of neurons in $L^{\mathrm{II}}$. Note that the change in the matrix $\mathbf{W}$ for this step is the learning rule in Equation 2.

$$\mathrm{W}_{\mu\nu}(\mathrm{t}_0 + \Delta \mathrm{t}) - \mathrm{W}_{\mu\nu}(\mathrm{t}_0) = (\eta \Delta \mathrm{t})(\mathrm{e}_\mu^{(\mathrm{i})}) \phi^{\mathrm{II}}(\mathrm{a}_\nu^{(\mathrm{i})}(\mathrm{t}_0)) - (\delta \Delta \mathrm{t}) \mathrm{W}_{\mu\nu}(\mathrm{t}_0) \tag{10}$$

From (10) and (8) we derive Equation 11 which gives the change in the weighted component $\gamma_\nu^{(q)}$ for some query stimulus $q$ over a single time step. We omit the subscript $\nu$ for brevity. The response of $L^{\mathrm{II}}$ to $q$ is the latent representation derived from the memory during the glimpse sequence.

$$\begin{aligned}
\gamma^{(q)}(t_0 + \Delta t) - \gamma^{(q)}(t_0) &= \sum_\mu (\mathrm{W}_{\mu\nu}(\mathrm{t}_0 + \Delta \mathrm{t}) - \mathrm{W}_{\mu\nu}(\mathrm{t}_0)) \mathrm{e}_\mu^{(\mathrm{q})} \\
&= (\eta \Delta t) \phi^{\mathrm{II}}(a^{(i)}(t_0)) \cdot \sum_\mu (e_\mu^{(i)} e_\mu^{(q)}) - (\delta \Delta t) \cdot \sum_\mu \mathrm{W}_{\mu\nu}(\mathrm{t}_0) \mathrm{e}_\mu^{(\mathrm{q})} \\
&= (\eta \Delta t) \phi^{\mathrm{II}}(a^{(i)}(t_0)) m_{iq}^{\mathrm{I}} - (\delta \Delta t) \gamma^{(q)}(t_0)
\end{aligned} \tag{11}$$

The next step in Block et al. (1962) is to define some sequence over the set of possible stimuli to be presented in order. We deviate slightly here as our sequence is not drawn from a bounded set. Instead, as we have seen, each glimpse is a sample from the manifold space of affine transforms over the image. As such, in Equation 12 we generalise Equation 11 to represent the change in the query response for some arbitrary point, $n$, in the sequence, with stimulus $g_n$ at time $t + n\Delta t$. Summing over the whole sequence we obtain Equation 13. We can now obtain an expression for the gradient of Equation 13 by dividing by $N\Delta t$ in the limit of $\Delta t \to 0$ (Equation 14).

$$\gamma^{(q)}(t + (n+1)\Delta t) - \gamma^{(q)}(t + n\Delta t) =$$
$$(\eta\Delta t)\phi^{\mathrm{II}}(a^{(g_n)}(t + n\Delta t))m_{g_n q}^{\mathrm{I}} - (\delta\Delta t)\gamma^{(q)}(t + n\Delta t) \quad (12)$$

$$\gamma^{(q)}(t + (N+1)\Delta t) - \gamma^{(q)}(t) =$$
$$\sum_{n=0}^{N}[(\eta\Delta t)\phi^{\mathrm{II}}(a^{(g_n)}(t + n\Delta t))m_{g_n q}^{\mathrm{I}} - (\delta\Delta t)\gamma^{(q)}(t + n\Delta t)] \quad (13)$$

$$\frac{d\gamma^{(q)}}{dt} = \lim_{\Delta t \to 0}\left[\frac{\gamma^{(q)}(t + (N+1)\Delta t) - \gamma^{(q)}(t)}{N\Delta t}\right]$$

$$= \eta\sum_{n=0}^{N}[\phi^{\mathrm{II}}(a^{(g_n)}(t))m_{g_n q}^{\mathrm{I}}] - \delta\gamma^{(q)}(t) \quad (14)$$

Now consider an iterative form of the equilibrium of Equation 14. Starting with $\gamma_0^{(g_n)} = 0$, $\forall n$ as an initial condition, we can obtain new values for each $\gamma^{(g_i)}$ and iterate by putting these back into the right hand side of Equation 15. We can show that this process converges under certain conditions. Specifically, if the right hand side of Equation 15 is nondecreasing and $\phi^{\mathrm{I}}$ and $\phi^{\mathrm{II}}$ are nonnegative and bounded by $\Phi^{\mathrm{I}}$ and $\Phi^{\mathrm{II}}$, successive values for $\gamma^{(g_i)}$ (and subsequently, responses $\phi^{\mathrm{II}}$) cannot decrease. A solution is found if at some step, no $\phi^{\mathrm{II}}$ increases. Alternatively, as shown in Equation 16, if values continue to increase then $\phi^{\mathrm{II}}$ approaches its upper bound, as $\chi$ approaches infinity.

$$\gamma_{\chi+1}^{(g_i)} = \frac{\eta}{\delta}\sum_{n=0}^{N}[\phi^{\mathrm{II}}(\beta^{(g_n)} + \gamma_\chi^{(g_n)})m_{g_n g_i}^{\mathrm{I}}] \quad (15)$$

$$0 \leq \lim_{\chi \to \infty}\gamma^{(g_i)} \leq \frac{\eta}{\delta}N^2(\Phi^{\mathrm{I}})^2\Phi^{\mathrm{II}} < \infty \quad (16)$$

Although Equation 14 is only an approximation of the dynamics of the system, we have demonstrated stability under certain conditions: firstly, the input to the memory network must not vary with time. This is satisfied in the case where the feature vector is not dependent on the output of a recurrent network. It is possible to extend this proof to incorporate such networks (Rosenblatt, 1962), however, that is outside the scope of this paper. Secondly, the activation functions must be nonnegative and have an upper bound (as with ReLU6 or Sigmoid). This introduces an interesting similarity as there is an upper bound to the number of times a real neuron can fire in a given window, governed by its refractory period. Finally, we can observe that Equation 15 is nondecreasing iff $\eta$ is greater than $\delta$, $\delta$ is positive and $\eta$ and $\theta$ are nonnegative.

## B  GLIMPSE SEQUENCE KL DIVERGENCE

The section details the derivation of the joint KL divergence term for our glimpse sequence, adapted from a derivation given in Dupont (2018). For a sequence of glimpse sub-spaces with $K$ components, $G = (\mathbf{g}_n)_{n=0}^{N}$, $\mathbf{g}_n \in \mathbb{R}^K$, we now derive a term for the KL divergence between the posterior and the prior for the joint distribution of the glimpses, $D_{\mathrm{KL}}(q_\phi(\mathbf{g}_0, \ldots, \mathbf{g}_N \,|\, \mathbf{x}) \,\|\, p(\mathbf{g}_0, \ldots, \mathbf{g}_N))$. Assuming that elements of $G$ are conditionally independent we have

$$q_\phi(G \,|\, \mathbf{x}) = \prod_{n=0}^{N} q_\phi(\mathbf{g}_n \,|\, \mathbf{x}) \ , \text{ and} \quad (17)$$

$$p(G) = \prod_{n=0}^{N} p(\mathbf{g}_n) \ . \quad (18)$$

We can therefore re-write the joint KL term and simplify

$$D_{\mathrm{KL}} \left( q_\phi(G \,|\, \mathbf{x}) \right) \| p(G)) = \mathbb{E}_{q_\phi(G \,|\, \mathbf{x})} \left[ \log \frac{q_\phi(G \,|\, \mathbf{x})}{p(G)} \right] \,, \tag{19}$$

$$= \mathbb{E}_{q_\phi(\mathbf{g}_0 \,|\, \mathbf{x}), \ldots, \mathbb{E}_{q_\phi(\mathbf{g}_N \,|\, \mathbf{x})}} \left[ \log \prod_{n=0}^{N} \frac{q_\phi(\mathbf{g}_n \,|\, \mathbf{x})}{p(\mathbf{g}_n)} \right] \,, \tag{20}$$

$$= \sum_{n=0}^{N} \mathbb{E}_{q_\phi(\mathbf{g}_0 \,|\, \mathbf{x}), \ldots, \mathbb{E}_{q_\phi(\mathbf{g}_N \,|\, \mathbf{x})}} \left[ \log \frac{q_\phi(\mathbf{g}_n \,|\, \mathbf{x})}{p(\mathbf{g}_n)} \right] \,. \tag{21}$$

Since $\mathbb{E}_{q_\phi(\mathbf{g}_i \,|\, \mathbf{x})}$, $\mathbf{g}_i \in G \setminus \{\mathbf{g}_n\}$ does not depend on $n$, we have

$$D_{\mathrm{KL}} \left( q_\phi(G \,|\, \mathbf{x}) \right) \| p(G)) = \sum_{n=0}^{N} \mathbb{E}_{q_\phi(\mathbf{g}_n \,|\, \mathbf{x})} \left[ \log \frac{q_\phi(\mathbf{g}_n \,|\, \mathbf{x})}{p(\mathbf{g}_n)} \right] \,, \tag{22}$$

$$= \sum_{n=0}^{N} D_{\mathrm{KL}} \left( q_\phi(\mathbf{g}_n \,|\, \mathbf{x}) \| p(\mathbf{g}_n) \right) \,. \tag{23}$$

For our experiments we set the prior to be an isotropic unit Gaussian ($p(\mathbf{z}) = \mathcal{N}(\mathbf{0}, \mathbf{I})$) yielding the KL term

$$D_{\mathrm{KL}} \left( q_\phi(\mathbf{z} \,|\, \mathbf{x}) \| \mathcal{N}(\mathbf{0}, \mathbf{I}) \right) = -\frac{1}{2} \left( 1 + \log \boldsymbol{\sigma}^2 - \boldsymbol{\mu}^2 - \boldsymbol{\sigma}^2 \right) \,, \tag{24}$$

and the joint KL term

$$D_{\mathrm{KL}} \left( q_\phi(G \,|\, \mathbf{x}) \| \mathcal{N}(\mathbf{0}, \mathbf{I}) \right) = -\frac{1}{2} \left( N + \sum_{n=0}^{N} \log \boldsymbol{\sigma}_n^2 - \sum_{n=0}^{N} \boldsymbol{\mu}_n^2 - \sum_{n=0}^{N} \boldsymbol{\sigma}_n^2 \right) \,, \tag{25}$$

where $(\boldsymbol{\mu}_n, \boldsymbol{\sigma}_n) = \mathbf{g}_n$. Note that in the case where each glimpse has the same prior, when taken in expectation over the latent dimensions, this summation is identical to concatenating the spaces.

## C    DRAWING STAGES

This Appendix gives the drawing stages diagrams for each of the drawing experiments. As discussed in the paper, there are multiple ways the model can learn to reconstruct line drawings such as those from the MNIST dataset. Figures 8a, 8b and 8c show the compression, parts based and the line drawing modes that are obtained for different glimpse sizes. Figure 9 shows the canvas sequence for CIFAR-10. Here, as discussed, the model is only able to compress the image with each glimpse and subsequently decompress with each sketch. Figure 10, however, shows the sketch sequence for the CelebA dataset, using the Bernoulli method to update the canvas. Here, the model has learned to transition smoothly from reconstructing low-information regions of the image to high-information regions resulting in a separation of background and foreground.

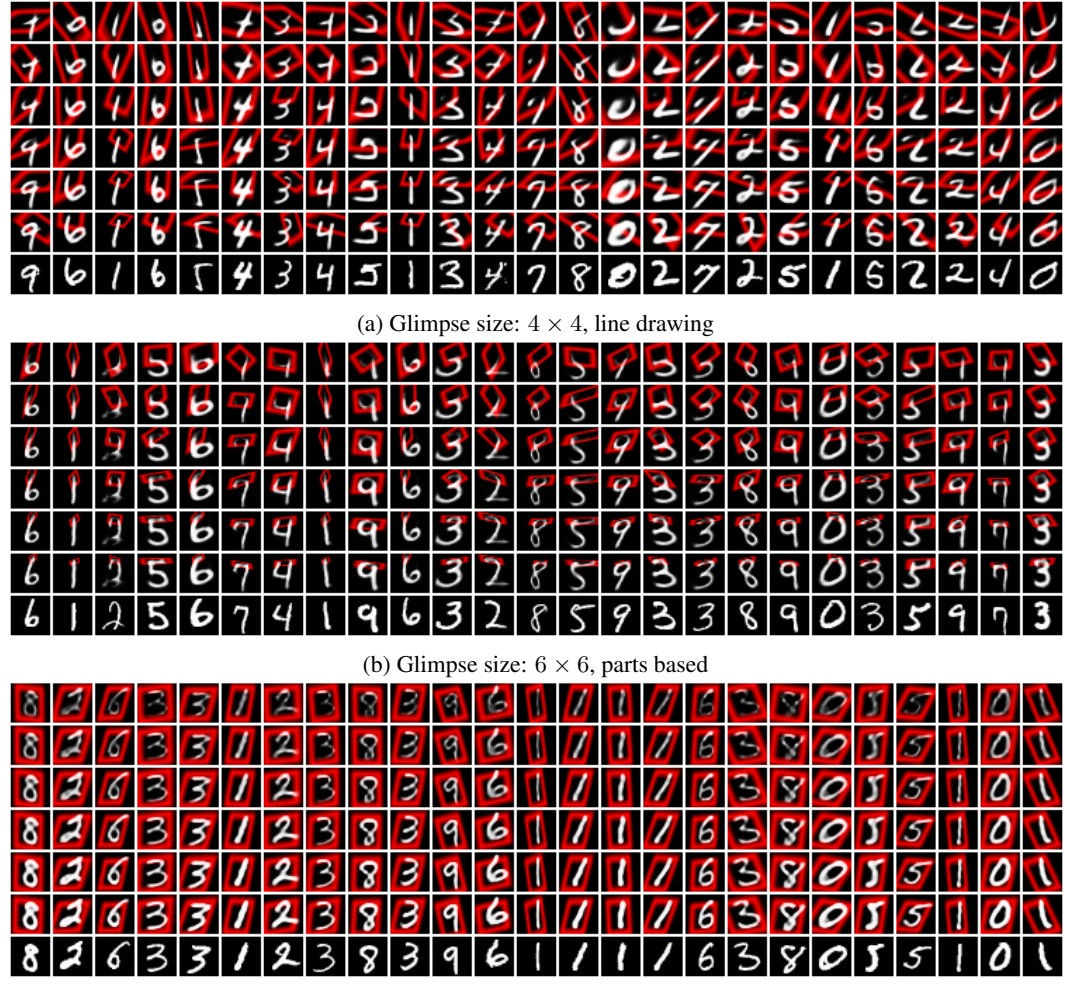

(a) Glimpse size: $4 \times 4$, line drawing

(b) Glimpse size: $6 \times 6$, parts based

(c) Glimpse size: $8 \times 8$, compression

Figure 8: Canvas updates for the drawing model on MNIST with 12 glimpses of size 8, 6 and 4. The first 6 rows give the drawing sequence (we omit the first 6 steps for brevity) and the bottom row shows the target image. Best viewed in colour.

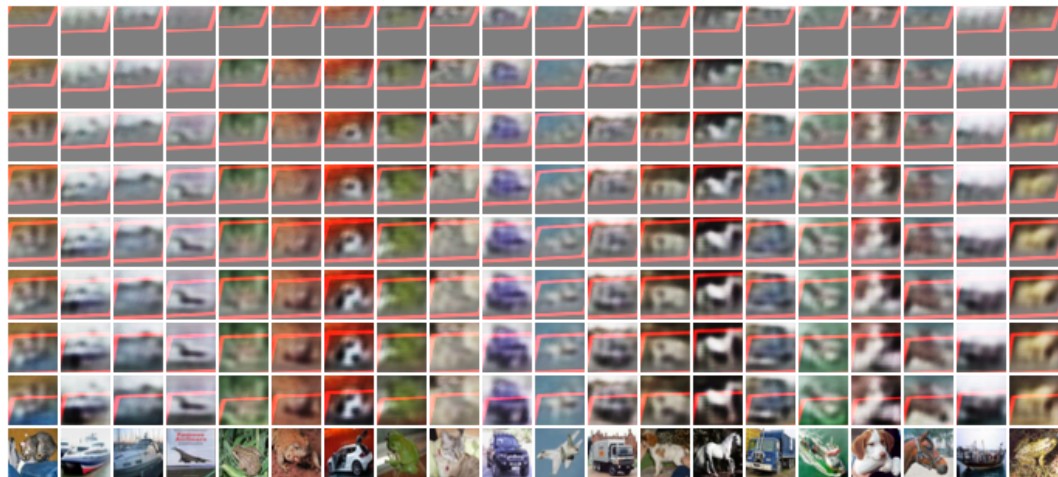

Figure 9: Canvas updates for the drawing model on CIFAR-10 with 8 glimpses of size 16. The first 8 rows give the drawing sequence and the bottom row shows the target image. Best viewed in colour.

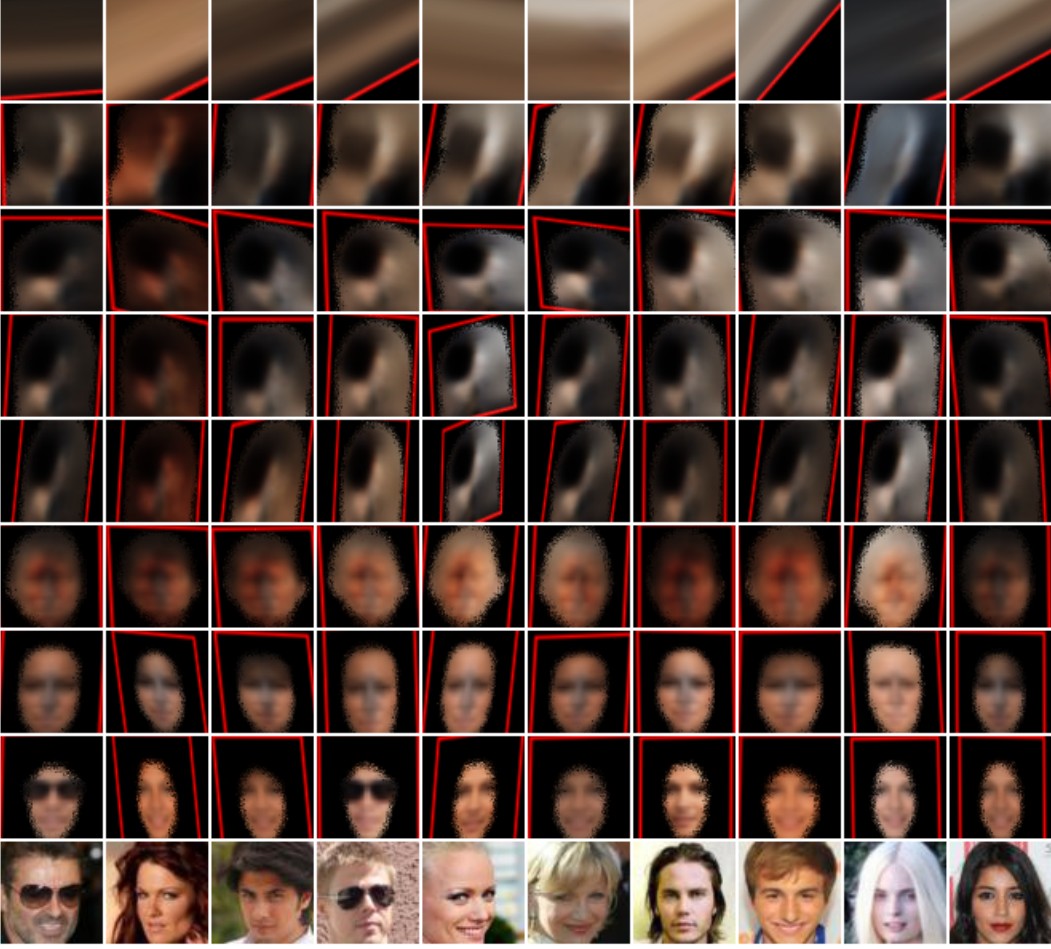

Figure 10: Canvas updates for the drawing model on CelebA with 8 glimpses of size 32. The first 8 rows give the drawing sequence and the bottom row shows the target image. Best viewed in colour.

# D    EXPERIMENTAL DETAILS

This Appendix details the specifics of the architectures used for the different experiments in the paper along with the training regimes. The code for these experiments can be found at `https://github.com/iclr2019-anon/STAWM`. Hyperparameters considered here include: the training scheme, data augmentations, data split and network architectures.

## D.1    MNIST

**All MNIST experiments:**

- Augmentation: random rotation $\pm 20°$
- Optimizer: Adam(lr $= 0.001$)
- Batch size: 128
- Dropout: 0.5
- Gradient clipping: 5
- RNN hidden size: $M \times 2$
- Context CNN:

    Layer 1: 64 Filters, $3 \times 3$, stride=2
    Layer 2: 128 Filters, $3 \times 3$, stride=2
    Layer 3: 256 Filters, $3 \times 3$, stride=2

- Query projection:

    Layer 1: Linear, context CNN output size $\times M$, ReLU6

- Classifier (if used):

    Layer 1: Linear, $M \times 10$, LogSoftmax

- Transpose CNN (if used):

    Transpose CNN with same structure as the glimpse CNN in reverse

**Classification $8 \times 8$:**

- Epochs: 200
- LR schedule: exponential of 0.99 and divided by ten at epochs $[50, 100, 150, 190, 195]$
- Memory size ($M$): 256
- Glimpse CNN:

    Layer 1: 64 Filters, $3 \times 3$
    Layer 2: 128 Filters, $3 \times 3$, stride=2

**Classification $28 \times 28$:**

- Epochs: 200
- LR schedule: exponential of 0.99 and divided by ten at epochs $[50, 100, 150, 190, 195]$
- Memory size ($M$): 512
- Glimpse CNN:

    Layer 1: 64 Filters, $3 \times 3$, stride=2
    Layer 2: 128 Filters, $3 \times 3$, stride=2
    Layer 3: 256 Filters, $3 \times 3$, stride=2

**Drawing $4 \times 4$:**

- Epochs: 100
- LR schedule: exponential of 0.99
- Memory size ($M$): 256
- Latent space: 4
- $\beta$: 4
- Glimpse CNN:

    Layer 1: 128 Filters, $3 \times 3$

**Drawing $6 \times 6$:**

- Epochs: 100
- LR schedule: exponential of 0.99
- Memory size ($M$): 256
- Latent space: 4
- $\beta$: 4
- Glimpse CNN:

    Layer 1: 128 Filters, $3 \times 3$

**Drawing $8 \times 8$:**

- Epochs: 100
- LR schedule: exponential of 0.99
- Memory size ($M$): 256
- Latent space: 4
- $\beta$: 4
- Glimpse CNN:

    Layer 1: 64 Filters, $3 \times 3$
    Layer 2: 128 Filters, $3 \times 3$, stride=2

**Self-supervised:**

- Epochs: 100
- LR schedule: exponential of 0.99
- Memory size ($M$): 256
- Glimpse CNN: same as $6 \times 6$ drawing mode

**Visual Sketchpad:**

- Epochs: 20
- LR schedule: exponential of 0.99
- Memory size ($M$): 256
- Latent space: 4
- $\beta$: 4
- Glimpse CNN: same as $6 \times 6$ drawing mode

### D.2 CIFAR-10

**All CIFAR-10 experiments:**

- Augmentation: random crop, random flip
- Batch size: 128
- Dropout: 0.5
- Gradient clipping: 5
- Query projection:
    Layer 1: Linear, context CNN output size $\times M$, ReLU6
- Classifier (if used):
    Layer 1: Linear, $M \times 10$, LogSoftmax
- Transpose CNN (if used):
    Transpose CNN with same structure as the glimpse CNN in reverse

**Classification with MobileNetV2** $32 \times 32$:

- Epochs: 300
- Optimizer: Adam(lr = 0.001)
- LR schedule: exponential of 0.99 and divided by ten at epochs $[150, 250]$
- Memory size: 1024
- RNN hidden size: 2048
- Context CNN:
    Layer 1: 64 Filters, $3 \times 3$, stride=2
    Layer 2: 128 Filters, $3 \times 3$, stride=2
    Layer 3: 256 Filters, $3 \times 3$, stride=2
- Glimpse CNN: MobileNetV2

**Drawing** $16 \times 16$:

- Augmentation: As above and colour jitter (25%)
- Epochs: 100
- Optimizer: Adam(lr = 0.0001)
- LR schedule: divide by 10 at epochs $[50, 90]$
- Memory size: 256
- Latent space: 32
- $\beta$: 2
- RNN hidden size: 256
- Context CNN:
    Layer 1: 32 Filters, $4 \times 4$, stride=2
    Layer 2: 32 Filters, $4 \times 4$, stride=2
    Layer 3: 32 Filters, $4 \times 4$, stride=2
- Glimpse CNN:
    Layer 1: 32 Filters, $4 \times 4$
    Layer 2: 64 Filters, $4 \times 4$, stride=2
    Layer 3: 128 Filters, $4 \times 4$, stride=2
    Layer 4: 128 Filters, $4 \times 4$, stride=2
- CNN initialisation: Kaiming uniform

**Baseline $\beta$-VAE:**

- Augmentation: As above and colour jitter (25%)
- Epochs: 100
- Optimizer: Adam(lr = 0.0005)
- LR schedule: divide by 10 at epochs [50, 90]
- Latent space: 32
- $\beta$: 2
- Glimpse CNN:

    Layer 1: 32 Filters, $4 \times 4$
    Layer 2: 64 Filters, $4 \times 4$, stride=2
    Layer 3: 128 Filters, $4 \times 4$, stride=2
    Layer 4: 128 Filters, $4 \times 4$, stride=2

- CNN initialisation: Kaiming uniform

**STAWM Self-supervised:**

- Epochs: 50
- Optimizer: Adam(lr = 0.001)
- LR schedule: divide by 10 at epochs [25, 40, 45]
- Classifier:

    Layer 1: Linear, $M \times 10$, LogSoftmax

**$\beta$-VAE Self-supervised:**

- Epochs: 50
- Optimizer: Adam(lr = 0.001)
- LR schedule: divide by 10 at epochs [25, 40, 45]
- Classifier:

    Layer 1: Linear, $32 \times 10$, LogSoftmax

### D.3 CELEBA

**Drawing $32 \times 32$:**

- Epochs: 100
- LR schedule: none
- Memory size: 256
- Latent space: 4
- $\beta$: 2
- RNN hidden size: 256
- Augmentation: none
- Optimizer: Adam(lr = 0.0001)
- Batch Size: 128
- Dropout: 0.3
- Gradient clipping: 5
- Context CNN:

    Layer 1: 32 Filters, $4 \times 4$, stride=2

Layer 2: 32 Filters, $4 \times 4$, stride=2
Layer 3: 64 Filters, $4 \times 4$, stride=2
Layer 4: 64 Filters, $4 \times 4$, stride=2

- Glimpse CNN:

    Layer 1: 32 Filters, $4 \times 4$
    Layer 2: 32 Filters, $4 \times 4$, stride=2
    Layer 3: 64 Filters, $4 \times 4$, stride=2
    Layer 4: 64 Filters, $4 \times 4$, stride=2

- Query projection:

    Layer 1: Linear, context CNN output size $\times M$, ReLU6

- Transpose CNN:

    Transpose CNN with same structure as the glimpse CNN in reverse

- CNN initialisation: Kaiming uniform

