# OpenReview forum: "A Biologically Inspired Visual Working Memory for Deep Networks"
_ICLR.cc/2019/Conference_

### Official Review · AnonReviewer1 · 2018-11-02
**Review for "A Biologically Inspired Visual Working Memory for Deep Networks" (Strong Accept after Revision)**

**Rating:** 9
**Confidence:** 5

**Review:**

Revision:

The authors have took in the feedback from myself and the other reviewers wholeheartedly, and have clearly worked hard to improve the results, and the paper during the revision process. In addition, their code release encourages easy reproducibility of their model, which imo is needed for this work given the non-conventional nature of the model (that being said, the paper itself is well written and the authors have done sufficiently well in explaining their approach, and also the motivation behind it, as per my original review). The code is relatively clear and self-contained demonstrating their experiments on MNIST, CelebA demonstrating the use of the visual sketch model.

I believe the improvements, especially given the compute resources available to the authors, warrant a strong accept of this work, so I revised my score to 9. I also believe this work will be of value to the ICLR community as it offers alternate, less explored approaches compared to methods that are typically used in this domain. I'm excited to see more in the community explore biologically inspired approaches to generative models, and I think this work along with the code base will be an important base for other researchers to use as a reference point for future work.

Original Review below:

Summary: They propose a biologically motivated short term attentive working memory (STAWM) generative model for images. The architecture is based on Hebbian Learning (i.e. associative memories are represented in the weight matrices that are dynamically updated during inference by a modified version of Hebbian learning rule). These memories are sampled from glimpses on an input image (using attention on contextual states, similar to [1]), in addition to a latent, query state. This model learns a representation of images that can be used for sequential reconstruction (via a sequence of updates, like a sketchpad, like DRAW [1], trained in an unsupervised manner). These memories produced by drawing can also be used for semi-supervised classification (achieves very respectable and competitive results for MNIST and CIFAR-10).

This paper is beautifully written, and the biological inspiration, motivation behind this work, and links to neuroscience literature as well as relation to existing ML work (even recent papers) is well stated. The main strength of this paper is that the author went from a biologically inspired idea to a complete realization of the idea in algorithmic form. The semi-supervised classification results are competitive to SOTA, and although the CIFAR-10 reconstruction results are not great (especially compared to generative adversarial models or recent variation models [2]), I think the approach is coming from a very different angle that is different enough compared to the literature to warrant some attention, or at least a glimpse, so to speak, from the broader community. The method may offer new ways to interpret ML models that is current models lack, which in itself is an important contribution. That being said, the fact that most adversarial generative models achieved a far better performance raises concern on the generalization ability of these memory-inspired learned representations, and I look forward to seeing future work investigate this area in more detail.

The authors also took great care in writing details for important parts of the experiments in the Appendix section, and open sourced the implementation to reproduce all their experiments. Given the complex nature of this model, they did a great job in writing a clear explanation, and provided enough details for the community to build biologically inspired models for deep networks. Even without the code, I felt I might have been able to implement most of the model given the detail and clarity of the writing, so having both available is a great contribution.

I highly recommend this paper for acceptance, with a score of 8 (edit: revised to 9 after rebuttal period). The paper might warrant a score of 9 if they had also achieved higher quality results for image generation, on Celeb-A or demonstrated results on ImageNet, and provided more detailed analysis about drawbacks of their approach vs conventional generative models.

[1] https://arxiv.org/abs/1502.04623
[2] https://arxiv.org/abs/1807.03039

---

> ### Author Response · Authors · 2018-11-15
> **Response to Reviewer 1**
>
> First, we would like to thank the reviewer for their time and their insightful, constructive comments. We will begin by discussing the changes that have been made in response to concerns regarding reconstruction quality and will then go on to address our progress towards providing a more detailed analysis of the drawbacks of our approach.
> The reviewer rightly noted that the reconstruction quality is not in line with state of the art models such as Glow [1]. However, the reconstructions produced by Glow were a result of approximately 600 convolutional layers totalling 200M parameters trained on 40 GPUs. Conversely, our model has around 10M parameters, at most 12 convolutional layers and was trained on 1 GPU. We therefore assert that, although the reviewer is undeniably correct in their assessment, a more suitable baseline should be sought. To this end, we have re-run our CIFAR-10 experiments (obtaining marginally improved results) and used the model from beta-VAE [2] to provide baseline reconstructions which can be seen in Figure 5 of the revised paper. It is now clear that our model outperforms the comparable VAE (test mean squared error of 0.0083 vs 0.0113 for the baseline). We have also been able to obtain substantially improved results for CelebA, which can again be seen in the revised paper. These increases came about from a more careful choice of hyperparameters and can be seen in both the reconstructions and the sketch sequences.
>
> We will now discuss the analysis of the drawbacks of our approach. On reflection, this was certainly missing from our initial manuscript and is an addition we have thought about carefully to ensure that we provide a sufficient analysis. For this, we have produced both qualitative and quantitative assessments of our model compared to other similar architectures (specifically, VAEs and DRAW). In Section 3.3 we have added some text which gives a clear comparison to the DRAW network and some of the limitations of STAWM. We have also obtained results for the self-taught setting for our network and the baseline VAE showing the performance gained (around 8%) by constructing a representation using the Hebb-Rosenblatt memory.
>
> To summarise, we have produced substantially improved reconstructions for CelebA and marginally better results for CIFAR-10 through a more careful choice of hyper-parameters. We have trained a comparable baseline VAE and compared reconstructions and performance in the self-taught setting to provide a more informative and transparent analysis. Finally, we have added some exposition which gives a greater assessment of the advantages and limitations of our approach. Overall we feel that these changes have significantly improved the quality of the paper and thank the reviewer for their valuable contributions. We hope that the reviewer finds these changes to their liking and welcome any further queries or comments.
>
> Kind regards,
> The Authors
>
> [1] Kingma, D. P., & Dhariwal, P. (2018). Glow: Generative flow with invertible 1x1 convolutions. arXiv preprint arXiv:1807.03039.
> [2] Higgins, I., Matthey, L., Pal, A., Burgess, C., Glorot, X., Botvinick, M., ... & Lerchner, A. (2016). beta-vae: Learning basic visual concepts with a constrained variational framework.

---

### Official Review · AnonReviewer2 · 2018-11-02
**More understanding and experiments are needed to better motivate the model.**

**Rating:** 5
**Confidence:** 4

**Review:**

The paper proposes a novel Hebb-Rosenblatt working memory model to augment the recurrent attention model and achieves competitive results on the MNIST dataset. Some results on CIFAR and Celeb-A datasets are also shown. The code is released anonymously which substantiated the reproducibility of the results; however, I haven’t physically run the code to verify it.

Motivation: First, as much as I appreciate the research direction of combining recurrent attention models and working memory, the use of recurrent attention models is not well motivated throughout the article. It it of course biologically inspired, but the engineering benefit is not obvious. It has the benefit of model compression, using less parameters to process the attended region, yet pooling mechanisms can also achieve similar effect. It also has the benefit of model interpretability, but for vanilla feed-forward counterparts, it is also possible to visualize salient regions that impact the decisions. While feed-forward CNNs process all regions in parallel, sequential models will be much slower. The core question is the following, if the final task is just image classification, why is sequential processing necessary? The author should spend some text explaining why studying recurrent attention model + working memory is important. Ideally, if the author could consider tasks other than image classification, which could potentially highlight the need for sequential processing and working memory.

Notation clarity: The clarity of model notation could be significantly improved. I am confused what is `e`, and my guess it is the layer I input. Figure 1 does not help my understanding. Notations such as \mu, \nu, M are left unexplained.

Understanding of the model: Although Hebbian learning rule is a well established mechanism, the article doesn’t provide much insight into why the rule is applied here but rather just stipulates them in Section 3.1 as the overall formulation of the memory network. What would be the objective function for such an update rule (e.g. the delta decay term corresponds to a L2 weight decay term)? For the applications studied in the paper, i.e. image classification, why does the model need to ever forget?

CIFAR/Celeb-A experiments: There is no tabulated results for these experiments. I only see CIFAR gets 93.05% accuracy, without mention of any baselines. Since attention mechanisms are used here, it would be a much stronger argument to report results on higher resolution images than CIFAR (which is 32x32).

Model interpretability: Model interpretability is often one of the biggest selling point of attention-based models. However, by examining the glimpse locations on CIFAR datasets, the model learns to look at the whole image for all glimpses, which hurts the argument of using a recurrent attention model. Also for MNIST experiments, the best number is achieved by using S_g=28, which has the glimpse size of the whole image. The author claims that it can learn a good representation despite the not so good looking glimpse visualization, but so can regular feed-forward CNNs.

Comparison to VAEs: Since VAE formulations are used, it would be good if the authors can compare the model with vanilla VAE and convolutional VAEs, both in terms of classification accuracy and reconstructed sample quality.

Comparison to DRAW: It is also recommended to show more visualization comparison to DRAW, and pinpoint the differences of using a canvas based memory vs. the proposed Hebb-Rosenblatt working memory.

In conclusion, I think the paper opens a promising direction of combining recurrent attention models and working memory networks. I believe it is a huge amount of engineering effort to make this model to work, as we can see in the Appendix and the released code base. However, there are several issues as I pointed above, most notably the motivation, and understanding of the models. The experiments on CIFAR and Celeb-A could been done more thoroughly, and I also believe that it would make more impact if the authors can show experiments other than image classification that highlight the need for a recurrent attention model equipped with working memory. Based on the above reasons, I think the paper could be better polished for future publication.

---

> ### Author Response · Authors · 2018-11-15
> **Response to Reviewer 2 (1/2)**
>
> To begin, we would like to thank the reviewer for their time and detailed comments. We will structure our response as follows. First, we will provide a detailed response to concerns relating to the motivation of the work. We will then go on to discuss steps that have been taken to provide a quantitative assessment of our model on CIFAR-10. Finally, we will address the remaining, minor comments by the reviewer.
>
> Regarding the motivation, the Reviewers 2 and 3 felt an absence of an appropriate reason given for undertaking this work in the way that we have. This is a concern that we have taken very seriously as it is central to our paper. We will now therefore devote some time to clarifying our motivation here, and discussing changes that have been made to the text. There were two key goals which motivated our approach. Firstly, we wished to understand if visual attention and working memory provide more than just increased efficiency and enable functions that cannot otherwise be achieved. Secondly, we wanted to understand and seek answers to some of the challenges faced when attempting to model such psychophysical concepts in deep networks. Throughout the paper we have taken steps towards achieving these goals. As noted by Reviewer 1 we move from 'a biologically inspired idea to a complete realization of the idea in algorithmic form'. Furthermore, we have demonstrated that Hebbian memories and visual attention can bring new capabilities to deep networks, such as multitasking and semantic segmentation. Evidently, we have failed to appropriately enunciate our intentions and, as suggested by the reviewer, we have devoted some text in the introduction and throughout the paper to more clearly stating and referencing these pivotal motivations.
>
> Regarding the CIFAR-10 experiments, the reviewer noted the absence of an appropriate baseline. The suggestion of comparing to VAEs is one that we wholeheartedly agree with, as such we have obtained reconstructions and self-taught results on CIFAR-10 for our model and a comparable beta-VAE [1]. These results show that STAWM outperforms this simple baseline in both cases, thereby demonstrating that our model is in line with expectations. Furthermore, by choosing a smaller latent space size, the new CIFAR-10 model learns to scan the image vertically, reconstructing along the way. The net effect of these experiments is to demonstrate that STAWM is learning something interesting that enables it to be more powerful than the static baseline.
>
> The reviewer noted an absence of any high resolution datasets from our experiments, with the largest being CelebA at 64 by 64. The problem of visual attention for large images is somewhat separate from our motivations and would consequently require too many amendments to the model to be included here, such as: foveal glimpses, a different memory rule and perhaps even a different attention mechanism. We do, however, agree that this is a fascinating potential direction for future research and have therefore elected to leave the challenge of higher resolution images as originally stated in our future work.
>
> Regarding the clarity of the notation, the reviewer is absolutely correct that \mu and \nu are not explained in the body of the paper. This was a mistake on our part as they are used in the appendix but do not need to be in the figure. They have therefore been removed. The other variables in the diagram, however, are discussed in Section 3.1 where the figure is referenced. We believe that this gives sufficient clarity but would happily consider adding a summary to the caption if the reviewer feels it is necessary.
>
> Regarding the understanding of the model, our main reason for using a Hebbian formulation is to maintain similarity with the psychophysical basis of our work as discussed above. We take some steps towards analysing the rule in the appendix section on stabilising the memory which shows that the decay term is necessary to prevent an explosion of the learning dynamics. Although the analogous ‘forgetting’ may not be useful for the task of classification (although this is highly dependant on context), it can be seen that it is of vital importance to the reconstruction setting. Furthermore, as these rates are learnable, an appropriate trade-off between learning and forgetting can be established during training. That said, more rigorous analysis of the memory would certainly aid understanding of the models performance and function. We, however, feel that to attempt to address this task in the setting of the paper would do it a disservice as we would be unable to reach sufficient depth or clarity in the limited space available. We have therefore elected to add this to our future work.

---

> ### Author Response · Authors · 2018-11-15
> **Response to Reviewer 2 (2/2)**
>
> Regarding the choice of glimpse size, the larger size on MNIST giving better results is something we entirely expected to observe since the model is able to learn a manifold representation over the pose space of the object, rather than its parts. This gives an alternate view of attentional models, which does not focus on efficiency but on increased performance and generalisation. We have added some text which explains this appropriately in Section 4.1.
> Both Reviewer 2 and Reviewer 1 noted a lack of discussion relating our model to DRAW. To address this, we have also taken the opportunity to add some exposition further relating our model to the DRAW network as suggested by the reviewer.
>
> Regarding the request for more experiments, we feel that the characterisation of our work as exclusively performing image classification is not entirely appropriate. During the course of the paper we experiment on a range of datasets with: classification, reconstruction, unsupervised feature learning, visual sketchpads and unsupervised segmentation. Furthermore, we perform these tasks with only minimal changes to the model, and go on to demonstrate that STAWM can be made to multitask in a way not previously studied.
>
> We will now give a brief summary of the changes we have made to the paper. Primarily, we have added some text giving an explicit reason for our interest in this approach. We have further been able to obtain better reconstruction results for CIFAR-10 and CelebA. We have also added self-taught results on CIFAR-10 and results from a comparable baseline VAE as suggested by the reviewer. We have then added some text which gives a more detailed comparison of our work with the DRAW network and discusses some of the limitations of our approach. Finally, we have added the recommendations by the reviewer which we feel are extensions beyond the scope of the paper to our future work. Overall, we feel that these changes have greatly improved the quality of the paper and we are extremely grateful to the reviewer for their assistance in this effort. To close, we hope that the reviewer finds these changes to their liking and welcome any further queries or comments.
>
> Kind regards,
> The Authors
>
> [1] Higgins, I., Matthey, L., Pal, A., Burgess, C., Glorot, X., Botvinick, M., ... & Lerchner, A. (2016). beta-vae: Learning basic visual concepts with a constrained variational framework.

---

### Official Review · AnonReviewer3 · 2018-11-03
**Interesting ideas, but muddled presentation.**

**Rating:** 4
**Confidence:** 4

**Review:**

Summary: this paper introduces a new  network architecture inspired by visual attentive working memory. The network consists of a recurrent components that generates glimpses of features from a CNN applied to the input (inspired by the DRAW network), and a working memory component that iterative stores memories using Hebbian mechanisms. The authors apply this network to classification tasks (MNIST) as well as using it as a generative model.

The ideas in the paper are somewhat  interesting, but I have major concerns with the motivation (it is unclear) and the experiments (not convincing):

Motivation: The authors motivate their inclusion of a Hebbian working memory from the perspective of trying to mimic the human visual system. The main problem here is that it is unclear what problem the authors are trying to solve by including this Hebbian mechanism. In the fast-weights paper (Ba et al), they had a clear example of a task that standard recurrent networks could not easily solve, which motivated the inclusion of a working memory mechanism. A similar motivation here is lacking, with the main justification seeming to be to "move towards a biologically motivated model of vision". Are the authors interested in more biologically motivated models because they think they will be useful for some task? Or are the authors interested in models of biological vision itself? If the former, it is unclear what new tasks would be solved by their model (all the results focus on tasks that can be solved without these mechanisms). If the latter, there should be some clear goals for what they hope their model to achieve. "Moving towards biological vision" is too vague and broad of a justification in order for us to judge progress. Section 2 discusses, at a high level, broad concepts from the visual neuroscience literature, but this also does not clearly motivate why the authors are interested in this particular instantiation of these ideas, indeed, their model is only weakly related to many of the neuroscience ideas discussed.

Results: The authors evaluate their network on two tasks: classification and image generation.
- For classification, I have a hard time understanding what these results tell us. Very simple models can achieve low test error on MNIST, so it is unclear what the attention or working memory buys you. One simple improvement would be if the authors ablated different parts of their network to show how critical they are to performance. Increasing the window size to 28 helps the model, suggesting that the network is hindered by the glimpses, so I do not feel like I have learned much from these results. In addition, the authors only mention Cifar10 results in a couple of lines, so it is hard to take anything away from those statements.
- For image generation, similarly, the authors do not compare their model to other standard generative models. Does their model perform better than simple baselines?
Finally, for all of these results, what is the working memory doing? Why is it necessary? Does it learn something interesting? It is hard to understand the significance of the work without answers to these questions.

---

> ### Author Response · Authors · 2018-11-15
> **Response to Reviewer 3**
>
> We thank the reviewer for their time and detailed comments regarding our paper. We will structure our response to echo the review, giving detailed comments for each of the points raised by the reviewer and stating any changes that have been made in response.
>
> Similar to Reviewer 2, Reviewer 3 notes issues with the motivation for our work. This is something that we have taken great consideration to address and will therefore devote some time to here. The motivation for our work is primarily to seek answers to some of the questions the reviewer poses. Specifically, we do not claim to know the benefit of a working memory, nor the advantage of visual attention. Instead, we look to discern if there is such an advantage to a working memory or sequential processing of an image. It is in this effort that we believe we have succeeded, as noted by Reviewer 1 we move from 'a biologically inspired idea to a complete realization of the idea in algorithmic form'. We consequently summarise two functions in our conclusion that come about from memory and attention that were not previously known: the ability to disentangle foreground from background and the ability to learn encoders which can produce a classification and a descriptive sketch that helps to interpret the model. Both of these functions are unique to our model and are only made possible by sequential processing and the presence of a memory. Such a motivation was evidently absent from the initial portion of the paper and so we have now added appropriate text to state this clearly as suggested by the reviewer.
>
> Regarding the related work, our approach is certainly only inspired by the contents of Section 2 rather than directly emulating them. However, many of the decisions we have made can be seen to potentially reduce performance but give a closer appropriation to the psychophysical foundation of our work. Towards the end of the paper, the advantages of this approach become clear as our model is able to naturally perform tasks that extend far beyond the scope of traditional architectures.
>
> Some concerns have been raised by Reviewers 2 and 3 regarding the effect of glimpse size and the demonstration that increasing the glimpse size improves performance. As noted above, our reasons for conducting experiments in the way that we have is to try to discern what the advantage of a visual memory is. In the case of the over-complete model, we expect that the memory can learn a positional manifold over the image by subjecting it to different transforms and observing the effect. With the two classification experiments we have concurrently demonstrated that the model can learn a complete representation from a series of parts and that it can learn an over-complete, manifold representation from a series of variants. We have added some exposition to the results section which clears this up.
>
> The reviewer noted an absence of quantitative assessment of the model on CIFAR-10 and comparison to a baseline. We have taken great care to address this by obtaining both reconstructions and self taught results for our model and a comparable beta-VAE [1] on CIFAR-10. These results now show that our model outperforms the simple baseline and is therefore in line with expectations.
>
> The final comments by the reviewer discuss the function of the memory. Analysing what the memory is doing is a subject which deserves to be addressed independently of its context and with rigorous mathematical analysis. We therefore feel that this is better suited as a statement for future work. We have, however, through experimentation taken some steps towards discussing the value of a memory. The memory can learn a context independent state which can subsequently be queried for a goal oriented representation. Furthermore, the memory can build up information in a fashion that enables semantic segmentation and a strong notion of object. Finally, the memory can facilitate networks which are able to effectively multi-task.
>
> To summarise, we have begun by giving a more complete statement of intent towards the beginning of the paper. We have then proceeded to obtain results for our model and a comparable baseline for the self taught setting on CIFAR-10. We have further added text which clarifies the choice of glimpse size in the MNIST classification experiments. Finally, we have stated the desire to understand more deeply the function of the memory rule in our future work. Overall, we feel that these changes have greatly improved the quality of the paper and thank the reviewer for their valued contribution. To close, we hope that the reviewer finds these changes to their liking and welcome any further queries or comments.
>
> Kind regards,
> The Authors
>
> [1] Higgins, I., Matthey, L., Pal, A., Burgess, C., Glorot, X., Botvinick, M., ... & Lerchner, A. (2016). beta-vae: Learning basic visual concepts with a constrained variational framework.

---

### Author Response · Authors · 2018-11-22
**Github Updated**

We have now updated the code on our Github ( https://github.com/iclr2019-anon/STAWM ) to include the new experiments from the rebuttal. We have also added some visualisations showing that the MNIST drawing model can learn a meaningful sequence even at 36 sketches. This is now in line with the current version of the paper.

Kind regards,
The authors

---

### Meta-Review · Area_Chair1 · 2018-12-11

**Confidence:** 2
**Recommendation:** Reject

**Metareview:**

The paper received mixed and divergent reviews. As a paper of unusual topic in ICLR, the presentation of this work would need improvement. For example, it is difficult to understand what's the overall objective function, why a specific design choice was made, etc. It's nice to see that the authors somehow did quite a bit of engineering to make their model work for classification and drawing tasks, but (as an ML person) it’s difficult to get a clear rationale on why the method works other than that it’s biologically motivated. In addition, the proposed model (at a functional level) looks quite similar to Mnih et al.'s "Recurrent Models of Visual Attention" work (for classification) and Gregor et al's DRAW model (for generation) in that all these models use sequential/recurrent attention/glimpse mechanisms, but no direct comparisons are made. For classification, the method achieves strong performance on MNIST but this may be due to a better architecture choice compared to Mnih's model but not due to the difference of the memory mechanism. For image generation/reconstruction, the proposed method seems to achieve quite good results but they are not as good as those from DRAW method. Overall, the paper is on the borderline, and while this work has some merits and might be of interest to some subset of audience in ICLR, there are many issues to be addressed in terms of presentation and comparisons. Please see reviews for other detailed comments.